# Impact of Intensive Care Unit Discharge Delay on Liver Transplantation Outcomes

**DOI:** 10.3390/jcm11092561

**Published:** 2022-05-02

**Authors:** Shirin Salimi, Keval Pandya, Rebecca Jane Davis, Michael Crawford, Carlo Pulitano, Simone Irene Strasser, Geoffrey William McCaughan, Avik Majumdar, Ken Liu

**Affiliations:** 1Australian National Liver Transplant Unit, Royal Prince Alfred Hospital, Sydney, NSW 2050, Australia; shirin.salimi1@gmail.com (S.S.); kevalvpandya@gmail.com (K.P.); michael.crawford1@health.nsw.gov.au (M.C.); carlo.pulitano@health.nsw.gov.au (C.P.); simone.strasser@health.nsw.gov.au (S.I.S.); g.mccaughan@centenary.org.au (G.W.M.); avik.majumdar@health.nsw.gov.au (A.M.); 2Department of Microbiology and Infectious Diseases, Royal Prince Alfred Hospital, Sydney, NSW 2050, Australia; rebecca.davis4@health.nsw.gov.au; 3Sydney Medical School, University of Sydney, Sydney, NSW 2006, Australia; 4Liver Injury and Cancer Program, The Centenary Institute, Sydney, NSW 2050, Australia

**Keywords:** liver transplantation, discharge delay, multi-resistant organism, bed-block, time-base targets

## Abstract

Background: For general intensive care unit (ICU) patients, ICU discharge delay (ICUDD) has been associated with an increased hospital length of stay (LOS) and the acquisition of multi-resistant organism (MRO) infections. The impact of ICUDD on liver transplant (LT) recipients is unknown. Methods: We retrospectively studied consecutive adult LT between 2011 and 2019. ICUDD was defined as >8 h between a patient being cleared for discharge to ward and the patient leaving the ICU. Results: 550 patients received LT and the majority (68.5%) experienced ICUDD. The median time between clearance for ward and the patient leaving the ICU was 25.6 h. No donor or recipient variables were associated with ICUDD. Patients cleared for discharge early in the week (Sunday-Tuesday) and those discharged outside routine work hours were more likely to experience ICUDD (*p* = 0.001 and *p* < 0.001, respectively). The median hospital LOS was identical (18 days, *p* = 0.96) and there were no differences in other patient outcomes. Patients who became colonized with MRO in the ICU spent a longer time there compared to those who remained MRO-free (9 vs. 6 days, *p* < 0.001); however, this was not due to ICUDD. Conclusion: ICUDD post-LT is common and does not prolong hospital LOS. ICUDD is not associated with adverse patient outcomes or MRO colonization.

## 1. Introduction

Liver transplantation (LT) is a life-saving treatment for select patients with severe liver disease and/or hepatocellular carcinoma. It is a major operation associated with morbidity and patients are often very unwell with decompensated cirrhosis or acute liver failure at the time of the transplant. The recipient’s physiologic reserve and non-liver comorbidities also factor into the complexities of peri-transplant care. Therefore, routine intensive care unit (ICU) admission post-operatively for optimal monitoring and management is recommended [1].

The decision to discharge a patient from the ICU to a hospital ward after an LT is a medical and surgical one based on the patient’s recovery, level of care available on the transplant ward, and complications from their underlying liver disease, comorbidities, or the LT itself. Despite medical clearance, timely discharge from the ICU can also be impeded by logistical obstacles, primarily a lack of ward bed availability [2]. Meeting time-based targets in other areas of the hospital such as the emergency department has been shown to be associated with reduced in-hospital mortality [3]. Among general ICU patients, discharge delay is associated with prolonged hospitalization and a greater risk of acquiring multi-resistant organisms (MRO) but with no significant difference in mortality [4,5,6]. Discharge delay from the ICU also places a significant financial burden on the healthcare system with an estimated cost of AUD 77 million (USD 57 million) per year in Australia [7]. However, the impact of ICU discharge delay (ICUDD) on LT recipients has not been specifically examined and previous studies of the general ICU population mostly involved non-LT centers. Therefore, we aimed to assess the prevalence, risk factors, and clinical impact of ICUDD after LT in our Australian quaternary-referral center.

## 2. Materials and Methods

### 2.1. Patients

A retrospective analysis was performed on consecutive adult deceased-donor LT recipients between July 2011 and June 2019 (8-year period) at a state LT referral center. All patients were transferred to the ICU post-operatively and then to a specialized transplant hospital ward for recovery until hospital discharge. Our ICU is a “closed” unit where management is led by the attending intensivist with guidance from the medical and surgical LT teams who review the patient on a daily basis. Eligibility for ICU discharge is determined on the ICU ward round that occurs at least three times a day with real-time documentation at the patient’s bedside. Once clearance for transfer is documented, the patient is waitlisted for a liver transplant ward bed and the bed flow manager is informed. Whilst awaiting exit from ICU, nursing care adjusts to reflect ward-based practices (standard 4-hourly observations with 2-hourly observations in the case of patient-controlled analgesia); however, medical assessments remain at three times per day (greater than the usual frequency on the transplant ward). ICUDD was defined as patient transfer out of ICU occurring >8 h after clearance by medical staff as documented in the medical record by the intensive care doctors or, in the absence of this, by the liver transplant team. The 8 h period was chosen based on previous studies of ICUs across Australia and New Zealand [2,6]. Patients were excluded if they died during their initial ICU admission. The study protocol was conducted according to the Declaration of Helsinki and was approved by the Sydney Local Health District Human Ethics Research Committee (RPAH Zone) with a waiver of informed consent (X19-0303).

### 2.2. Clinical Data

Patient demographic and clinical data results were obtained from a prospective LT database and electronic medical records. In patients who survived to discharge from ICU after LT, we compared the following outcomes between those who experienced ICUDD versus those who did not. The primary outcome of interest was the ward length of stay (LOS) defined as time in hospital after initial ICU discharge. Secondary endpoints included graft survival (time to re-transplantation or death), patient survival (time to death), hospital LOS (total days in hospital since the LT operation), total length of ICU stay, ward LOS (time in hospital after initial ICU discharge), unplanned ICU readmission (subsequent ICU admission within the same hospital stay), unplanned hospital readmission (unscheduled readmission to hospital within 30- and 90-days post-discharge), and new colonization with MRO. We also compared the above outcomes in patients who became newly colonized with MRO versus those who did not. At our center, all patients undergo a screening swab for methicillin-resistant *Staphylococcus aureus* (MRSA), vancomycin-resistant Enterococci (VRE), extended-spectrum β-lactamase (ESBL)-producing organisms, and carbapenem-resistant Enterobacteriaceae (CRE) on entry to and exit from ICU and every seven days in between. New MRO colonization was defined as an initial negative swab followed by a positive swab detected during the patient’s ICU admission or within seven days of arrival on the hospital ward. Standard working hours were defined as 0800 to 1700 and the working week was divided into early (Sunday to Tuesday) and late (Wednesday to Saturday) based on previous studies of variations in hospital occupancy [8,9].

### 2.3. Statistical Analysis

Continuous variables were expressed in mean ± standard deviation (SD) or median (interquartile range [IQR]) as appropriate. Differences between subgroups were analyzed using χ2 or Fisher exact test for categorical variables and Student’s t-test, Mann–Whitney test, or one-way ANOVA for continuous variables as appropriate. The Kaplan–Meier method with log-rank test was performed to estimate cumulative survival and determine statistical significance. Statistical analysis was performed by Statistical Package for Social Science (SPSS version 23.0, Armonk, NY, USA). A result was considered statistically significant if *p* ≤ 0.05.

## 3. Results

### 3.1. Patient Characteristics

During the study period, 565 patients received LT. Fifteen patients were excluded due to death during the initial ICU admission. A total of 550 patients were included in the final analysis. The patient clinical characteristics are presented in Table 1.

The median time between clearance for ward and the patient leaving the ICU was 25.6 h (IQR 6.6–38.6) for the entire cohort. ICUDD was experienced by the majority of patients (377, 68.5%) in comparison to 173 patients (31.5%) who did not experience ICUDD. In those with ICUDD, the median duration of delay was 30.7 h (IQR 24.5–52.6). An absence of ward bed availability contributed to delays for all ICUDD patients. In more detail, ward bed availability alone accounted for ICUDD in 365 patients (96.8%). The remaining 12 patients experienced additional delay contributors including staffing shortages (nurses and porters) for 9 patients (2.4%), awaiting discharge documentation for 4 patients (1.1%), and avoiding after-hours discharge from the ICU for 2 patients (0.5%). The proportion of patients with ICUDD fluctuated during the study period with the period 2014–2016 experiencing the greatest rate of ICUDD (74.5%, vs. 59.7% in the period 2011–2013 and 68.6% in the period 2017–2019, *p* = 0.013). The patients ready for discharge earlier in the week (Sunday to Tuesday) were more likely to experience ICUDD than those cleared later in the week (Wednesday to Saturday; 77.5% vs. 62.2%, odds ratio 1.85, 95% CI 1.28–2.67, *p* = 0.001). Patients cleared for discharge outside of routine working hours were more likely to experience ICUDD than those cleared within working hours (93.6% vs. 66.2%, odds ratio 0.13, 95% CI 0.04–0.44, *p* < 0.001). Patients who were already in the ICU prior to LT trended towards having a lower rate of ICUDD (12.7% vs. 19.1%, *p* = 0.051). No other donor and recipient variables were associated with ICUDD (*p* > 0.05 for all, Table 1).

### 3.2. Patient Outcomes

As expected, the median LOS in the ICU post-LT was significantly longer in patients who experienced ICUDD compared to those who did not for both the initial ICU admission (5 vs. 3 days, *p* < 0.001) and the total time spent in the ICU during the entire hospital admission post-LT (6 vs. 5 days, *p* < 0.001; Table 2). However, the median hospital LOS was the same for the two groups (18 days, *p* = 0.96). After the patients were discharged from their initial ICU admission, the ward LOS was thus significantly less in the ICUDD group (13 vs. 15 days, *p* = 0.020). There were no significant differences in the rates of unplanned ICU and hospital readmissions and MRO colonization between patients with and without ICUDD (*p* > 0.05 for all, Table 2). After a median follow-up period of 36 months (IQR 13–59 months), there were 54 deaths and 18 re-transplants in our cohort. By Kaplan–Meier analysis, graft and patient survival did not differ between the two groups (Log-rank *p* = 0.38 and 0.56, respectively; Figure 1).

Of the original study cohort, 451 patients (82.0%) had adequate MRO screening data, of whom 40/451 acquired a new MRO infection during their ICU initial admission (8.9%). The MRO acquired were MRSA (*n* = 2, 0.5%) and VRE (*n* = 38, 8.4%). There were no cases of ESBL or CRE acquisition. These patients had a longer initial ICU admission post-LT compared to those who did not acquire a new MRO (9 vs. 6 days, *p* < 0.001). However, this difference in time was not associated with ICUDD, which was similar for the two groups at 67.5% in those with newly colonized MRO vs. 69.8% in those without newly colonized MRO (*p* = 0.76). Conversely, the proportion of patients already colonized with MRO prior to their LT admission did not differ for those with and without ICUDD (24.2% vs. 27.3% colonized with MRO, respectively, *p* = 0.43). No recipient variables (male, age, discharge year, inpatient status prior to transplant) were associated with new MRO colonizations in the ICU (*p* > 0.05 for all; Table 3). New MRO colonizations in the ICU were associated with greater total hospital LOS (26 days vs. 18 days), total ICU LOS (6 vs. 9 days), and rate of unplanned ICU readmission (27.5% vs. 9.5%) compared to those who did not acquire new MRO (*p* < 0.01 for all). However, no differences were seen in graft and patient survival between the two groups (Log-rank *p* = 0.61 and 0.47, respectively; Figure 2).

## 4. Discussion

Post-LT care is resource-intense with all patients requiring ICU admission [10,11]. We present the first study to assess the prevalence and clinical impact of ICUDD in LT recipients. We followed 550 patients undergoing deceased donor LT and found the majority (68.4%) experienced ICUDD >8 h. Unsurprisingly, patients with ICUDD spent a longer time in the ICU; however, there were no statistically significant differences in total hospital LOS or patient outcomes including MRO colonization.

Our results demonstrate that ICUDD is primarily due to factors related to hospital bed management (96.8%) rather than donor, recipient, or operative factors. This is consistent with findings from previous studies of the general ICU population attributing 74–92% of ICUDD to a lack of ward bed availability [2,4,12]. We observed a higher proportion of ICUDD at our center compared to these previous studies (68% vs. 27–50%), which likely reflects the need for all our LT patients to step down to a specific transplant ward, whereas non-LT ICU patients may have the option of recovering in one of several general hospital wards [2,4]. The predictors of ICUDD in our study were discharge earlier in the week, and discharge outside of routine working hours, which are congruent with previous reports [2,4]. A single-center study of 652 ICU discharges similarly observed that discharges occurring between Saturday and Monday were more likely to experience ICUDD (2.17 times) compared to discharges occurring between Tuesday and Friday [2]. These authors also noted that rates of ICUDD increased proportionately as ICU bed occupancy (a reflection of hospital occupancy) increased from 40% to 80%. In a separate prospective study of 955 general ICU patients across five Australian hospitals, Tiruvoipati et al. found that after-hours discharges were three times more likely to result in ICUDD (34% vs. 10%) [4]. These predictors may point to times when free beds are likely to be occupied already. Indeed, previous studies have shown peak hospital bed occupancy early in the working week (Monday and Tuesday) [8,9]. Despite evidence showing that after-hours discharges may lead to an increased risk of readmission and death [13,14,15], this has not translated into poorer outcomes for patients with ICUDD [2,4]. Indeed, our study did not detect any statistically significant negative impact of ICUDD on post-LT outcomes including graft survival, patient survival, and unplanned readmission rates; however, larger multi-center studies are required to explore this further.

We also observed an identical total hospital LOS between patients who experienced ICUDD and those who did not. This suggests that the patient convalescence process (including regaining mobility and functional status, progression of diet, etc.), begins in the ICU and not only after a patient is discharged to the ward. Indeed, at our center, all post-LT patients are routinely seen by physiotherapists and dieticians while still in the ICU and are encouraged to sit out of bed and begin mobilizing as soon as appropriate. In contrast, the aforementioned study by Tiruvoipati et al. found a small but significant increase in the total hospital LOS by one day in patients with ICUDD compared to those without [4]. This difference was entirely due to the ICUDD time (median delay 24 h) since the median time spent in hospital after ICU discharge was identical for both groups (5 days). The authors proposed that a certain amount of time is required by the treating team to prepare a patient for discharge regardless of time spent in the ICU after the discharge decision. Our patients (regardless of ICUDD status) experienced a much longer median hospital LOS of 18 days versus five days in the Tiruvoipati et al. cohort [4]. However, patients who experienced ICUDD in our study also had significantly shorter stays in the LT ward after discharge from the ICU compared to those without ICUDD (13 vs. 15 days), again suggesting that some of the convalescences occurred in the ICU.

Regarding new MRO colonization, our results confirmed previous studies which showed that a prolonged time in the ICU and a prolonged total hospital LOS were associated with increased risk. However, this extra time did not result from ICUDD since the rate and duration of ICUDD between patients who did and did not acquire new MRO were similar. Instead, these patients remained in the ICU/hospital longer because they needed ongoing care suggesting they were patients who were sicker and/or who experienced more complications post-LT. Thus, it does not appear that new MRO colonization can be reduced by addressing ICUDD. Other risk factors for new MRO colonization in these patients include the need for broad-spectrum antibiotics, invasive procedures and catheters, and prolonged intubation [5], although these were not specifically examined in the present study. Although MRO infection during the ICU stay has been reported to be associated with increased mortality in the literature [16,17,18], our cohort did not demonstrate a difference in patient or graft survival, bringing into question the impact of postoperative MRO infection on long-term survival. Nonetheless, this serves as a reminder that all clinicians should practice good antibiotic stewardship, hand hygiene, and other infection control measures for all patients.

The main strength of our study lies in our large cohort of LT patients spanning multiple years. However, several limitations should be acknowledged. First, the retrospective nature of this study relies on the accuracy and completeness of data found in medical records. Second, other undesirable patient consequences from ICUDD reported in other studies such as delirium or sleep disturbance are not routinely captured in our LT database and could not be studied. Finally, this single-center study may not reflect the situation in other institutions. Indeed, each LT center has its own unique caseload, bed management procedures, and logistics that would determine ICUDD. However, our results were largely consistent with those found in the general ICU population in Australia [2,4]. Indeed, our results should be confirmed with larger multi-center studies.

## 5. Conclusions

In conclusion, ICUDD post-LT is common and is most likely due to logistical factors. The discharge delay was not associated with prolonged hospital LOS or adverse patient outcomes. Although prolonged time in the ICU was associated with an increased risk of new MRO colonization, this was not directly contributed to by ICUDD.

## Figures and Tables

**Figure 1 jcm-11-02561-f001:**
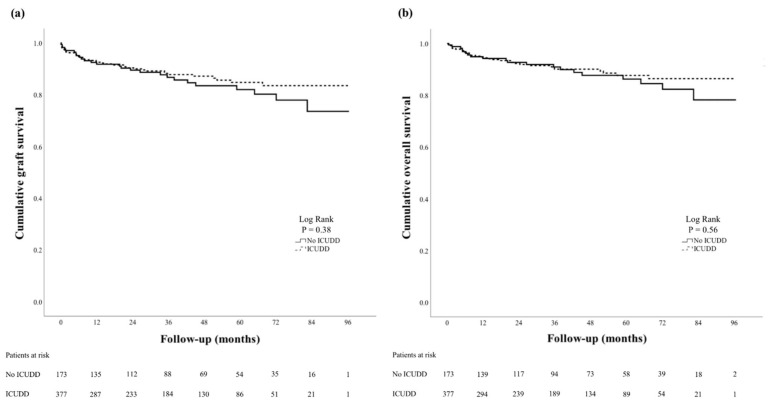
Survival analyses. Kaplan–Meier analyses of cumulative liver graft survival (**a**) and overall survival (**b**) in patients with no intensive care unit discharge delay (ICUDD) and patients with ICUDD.

**Figure 2 jcm-11-02561-f002:**
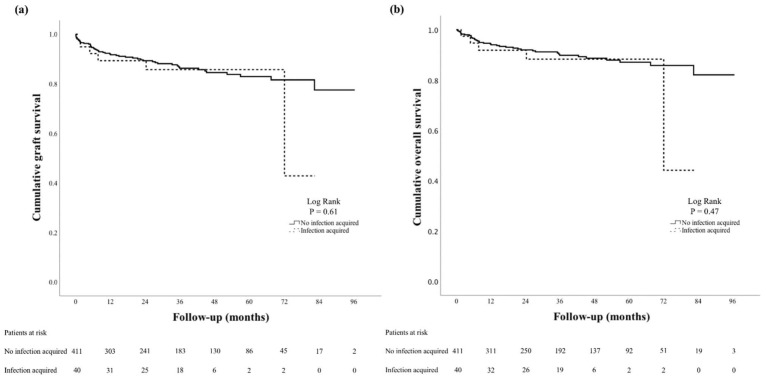
Survival analyses. Kaplan–Meier analyses of cumulative liver graft survival (**a**) and overall survival (**b**) in patients who did not become newly colonized with a multi-resistant organism infection and patients who did become colonized.

**Table 1 jcm-11-02561-t001:** Clinical characteristics of LT recipients with and without ICUDD.

Characteristic	All *n* = 550	No ICUDD *n* = 173	ICUDD *n* = 377	*p* Value
Male	386 (70.2)	128 (74.0)	258 (68.4)	0.19
Age (years)	54 (47–59)	53 (46–60)	54 (47–59)	0.63
Primary indication for LT				0.44
HCC	142 (25.8)	47 (27.2)	95 (25.2)	
Decompensated cirrhosis	378 (68.7)	117 (67.6)	261 (69.2)	
HCV	106 (19.3)	39 (22.5)	67 (17.8)	
Alcohol related liver disease	78 (14.2)	24 (13.9)	54 (14.3)	
Primary sclerosing cholangitis	44 (8.0)	12 (6.9)	32 (8.5)	
NAFLD	43 (7.8)	12 (6.9)	31 (8.2)	
Other	107 (19.4)	30 (17.4)	77 (20.4)	
Acute liver failure	30 (5.5)	9 (5.2)	21 (5.6)	
Retransplantation patient	22 (4.0)	9 (5.2)	13 (3.4)	0.33
Combined liver-kidney transplantation	21 (3.8)	6 (3.5)	15 (4.5)	0.77
Pre-transplant MELD score	19 (14–25)	19 (14–27)	19 (14–24)	0.59
DCD donor	40 (7.2)	11 (6.4)	29 (7.7)	0.58
DRI	1.6 (1.3–1.8)	1.6 (1.3–1.8)	1.6 (1.4–1.8)	0.89
ICU inpatient prior to transplant	81 (14.7)	33 (19.1)	48 (12.7)	0.051
MRO colonization prior to ICU admission	138 (25.2)	47 (27.3)	91 (24.2)	0.43
Discharge year				0.013 ^a^
2011–2013	144 (26.2)	58 (40.2)	86 (59.7)	
2014–2016	212 (38.5)	54 (25.6)	158 (74.5)	
2017–2019	194 (35.3)	61 (31.4)	133 (68.6)	
Discharge in early week	267 (48.5)	66 (38.2)	201 (53.3)	0.001
Discharge during weekday	429 (78.0)	141 (81.5)	288 (76.4)	0.18
Discharge within working hours	503 (91.5)	170 (98.3)	333 (88.3)	<0.001

The data are shown in number (percentage) and median (interquartile range). ^a^ *p*-value < 0.017 for 2011–2013 vs. 2014–2016. DCD, donation after circulatory determination of death; DRI, donor risk index; HCC, hepatocellular carcinoma; HCV, hepatitis C virus; ICU, intensive care unit; ICUDD, intensive care unit discharge delay; LT, liver transplantation; MELD, model for end-stage liver disease; MRO, multi-resistant organism; NAFLD, non-alcoholic fatty liver disease.

**Table 2 jcm-11-02561-t002:** Comparison of outcomes between patients with and without ICUDD.

Characteristic	No ICUDD *n* = 173	ICUDD *n* = 377	*p* Value
Ward LOS (days)	15 (10–22)	13 (8–20)	0.02
Total hospital LOS (days)	18 (13–29)	18 (13–27)	0.96
Initial ICU admission LOS (days)	3 (2–6)	5 (4–8)	<0.001
Total ICU LOS including readmissions (days)	5 (3–8)	6 (5–10)	<0.001
Unplanned return to operating theater	40 (23.1)	72 (19.1)	0.28
Unplanned ICU readmission	21 (12.1)	29 (7.7)	0.09
Unplanned hospital readmission within 30 days	55 (32.2)	121 (32.7)	0.90
Unplanned hospital readmission within 90 days	86 (50.6)	173 (46.9)	0.42
New colonization with MRO	13 (9.5)	27 (8.6)	0.76

The data are shown in number (percentage) and median (interquartile range). ICU, intensive care unit; ICUDD, intensive care unit discharge delay; LOS, length of stay; LT, liver transplantation; MRO, multi-resistant organism.

**Table 3 jcm-11-02561-t003:** Characteristics of patients who did vs. those who did not acquire MRO colonization during ICU.

Characteristic	No Infection Acquired *n* = 411	Infection Acquired *n* = 40	*p* Value
Male	292 (71.0)	26 (65.0)	0.42
Age (years)	55 (47–59)	57 (50–61)	0.09
Discharge year			0.011 ^a^
2011–2013	88 (97.8)	2 (2.2)	
2014–2016	161 (87.0)	24 (13.0)	
2017–2019	162 (92.0)	14 (8.0)	
Hospital inpatient prior to transplant ICU inpatient prior to transplant	188 (45.7) 55 (13.4)	14 (35.0) 5 (12.5)	0.78 0.88
ICUDD	287 (69.8%)	27 (67.5%)	0.76
Duration of delay in those with ICUDD (hours)	31 (24–53)	33 (28–74)	0.25
Total hospital LOS (days)	18 (13–27)	26 (15–49)	0.007
Initial ICU admission LOS (days)	4 (3–7)	7 (5–11)	<0.001
Ward LOS (days)	13 (8–21)	18 (10–30)	0.06
Total ICU LOS including readmissions (days)	6 (4–9)	9 (6–16)	<0.001
Unplanned return to operating theater	84 (20.4)	10 (25)	0.50
Unplanned ICU readmission	39 (9.5)	11 (27.5)	0.001
Unplanned hospital readmission within 30 days	130 (32.2)	12 (30.8)	0.86
Unplanned hospital readmission within 90 days	196 (48.8)	17 (43.6)	0.54

The data are shown in number (percentage) and median (interquartile range). ^a^ *p*-value < 0.017 for 2011–2013 vs. 2014–2016. ICU, intensive care unit; ICUDD, intensive care unit discharge delay; LOS, length of stay; LT, liver transplantation; MRO, multi-resistant organism.

## Data Availability

Supporting data can be made available on reasonable request.

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
