# Peer review of "Impact of Intensive Care Unit Discharge Delay on Liver Transplantation Outcomes"

_jcm, 2022, doi:10.3390/jcm11092561_

Round 1
Reviewer 1 Report
Good work and good manuscript. few points can be improved.
Regarding title:
Long title. Can be improved. Can be changed to: "Delay of ICU discharge and its impact on liver transplantation outcome"
Abstract: line 14 the word "recipients" should be put after LT.
Introduction: line 36; it is mentioned that the decision to discharge is a medical one. In the situation of liver transplantation; the decision to discharge is medical and surgical. so, this needs correction.
Methodology: line 77, it is mentioned that the primary outcome was the number of days of total H stay since the operation. How this can serve the aim mentioned previously? should be corrected to the number of H stay after ICU discharge.
Also, the authors should clearly divide recipients into 2 groups and compare between them:
G1= ICUDD=377 recipients
G2=No ICUDD=173 recipients
Results and discussion:
Table 1 shows the demographics of the patients. Is the age mean or average?
In table 2; The difference in the unplanned H readmission, unplanned ICU readmission, and infection is higher in the ICUDD group although the difference was not statistically significant. Authors did not mention this although it is a positive results but need further work involving higher number of recipients.
Fig 1 has no legend
Line 160 mentions that "No donor or recipient variables were associated with new MRO colonization". Although, no variables were included in the methods, results, tables or abstract.!!
The paragraph from line 152 to 165 seems not in place.
Table 3 shows results of MRO patients (411+40= 451). This number was not mentioned before. I assume it is the number of recipients with MRO. This should be clarified.
Regarding discussion:
In the first paragraph; line 180 state that authors follow 550 patients!. There was no mention of follow up before in the whole manuscript, was there? and for how long??
in the same paragraph; there is a mention that there were no difference in outcome on all levels between the 2 groups. It is better to interpret the results as the outcome was worse in the ICUDD group but the differences were not statistically sig.
Line 238: there is an explanation for the unexpected results in this regard; that the impact of early/immediate (usually nosocomial) postoperative infections on long term survival is little if any.
Author Response
Thank you for your valuable feedback on our paper. Please find a point by point response below.
Long title. Can be improved. Can be changed to: "Delay of ICU discharge and its impact on liver transplantation outcome"
- The title has been shortened to “Impact of intensive care unit discharge delay on liver transplantation outcomes”.
Abstract: line 14 the word "recipients" should be put after LT.
- The word “recipients” has been placed after “(LT)” in line 13.
Introduction: line 36; it is mentioned that the decision to discharge is a medical one. In the situation of liver transplantation; the decision to discharge is medical and surgical. so, this needs correction.
- Lines 36-37 have been updated to include decisions are based on medical and surgical opinion.
Methodology: line 77, it is mentioned that the primary outcome was the number of days of total H stay since the operation. How this can serve the aim mentioned previously? should be corrected to the number of H stay after ICU discharge.
- The primary outcome has been amended to ward length of stay, defined as time in hospital after initial ICU discharge. The hospital length of stay has been changed to a secondary outcome (lines 81-84). Table 2 has also been updated to reflect this change.
Also, the authors should clearly divide recipients into 2 groups and compare between them:
G1= ICUDD=377 recipients
G2=No ICUDD=173 recipients
- The results section has been updated to more clearly outline the two groups being compared in the study (line 115). The results, tables, figures and discussion primarily compare patients with and without ICUDD as depicted in the table title and column headings.
Results and discussion:
Table 1 shows the demographics of the patients. Is the age mean or average?
- The legend of table 1 has been updated to indicate the age given is the median age.
In table 2; The difference in the unplanned H readmission, unplanned ICU readmission, and infection is higher in the ICUDD group although the difference was not statistically significant. Authors did not mention this although it is a positive results but need further work involving higher number of recipients.
- In regards to directly comparing the percentage outcomes in the ICUDD group vs non-ICUDD group, both groups are either very similar or outcomes appear better in the ICUDD group. Lines 226-229 in the discussion have been updated to emphasise the lack of statistical significance and the importance of further studies in larger cohorts.
Fig 1 has no legend
- We apologise for this oversight, the figure 1 legend has been added.
Line 160 mentions that "No donor or recipient variables were associated with new MRO colonization". Although, no variables were included in the methods, results, tables or abstract.!
- This sentences has been corrected to only include the recipient variables outlined in table 3, namely age, gender, and inpatient status prior to transplant (lines 177-178).
The paragraph from line 152 to 165 seems not in place.
- The formatting and placement of this paragraph has been corrected.
Table 3 shows results of MRO patients (411+40= 451). This number was not mentioned before. I assume it is the number of recipients with MRO. This should be clarified.
- Of the original study cohort, 451 patients had sufficient MRO screening data for analysis. This clarification has been included in the manuscript (line 168).
Regarding discussion:
In the first paragraph; line 180 state that authors follow 550 patients!. There was no mention of follow up before in the whole manuscript, was there? and for how long??
- The follow-up period of 36 months is stated in line 152 of the results section. The total number of patients in the cohort is stated in line 114 of the results section.
in the same paragraph; there is a mention that there were no difference in outcome on all levels between the 2 groups. It is better to interpret the results as the outcome was worse in the ICUDD group but the differences were not statistically sig.
- In regards to comparing the percentage outcomes in the ICUDD group vs non-ICUDD group, both groups are either very similar or outcomes appear better in the ICUDD group. Lines 202 and 226-229 have been amended to clarify our results did not reach statistical significance.
Line 238: there is an explanation for the unexpected results in this regard; that the impact of early/immediate (usually nosocomial) postoperative infections on long term survival is little if any.
- Lines 262-263 in the discussion have been updated to reflect this point.
Reviewer 2 Report
The conclusion seems to stop suddenly. The author should finish it.
Author Response
Thank you for your valuable feedback on our paper.
The conclusion seems to stop suddenly. The author should finish it.
- We apologise for this oversight, the sentence has been corrected (lines 280-281).
This manuscript is a resubmission of an earlier submission. The following is a list of the peer review reports and author responses from that submission.